# Modelling Virgin Olive Oil Potential Shelf-Life from Antioxidants and Lipid Oxidation Progress

**DOI:** 10.3390/antiox11030539

**Published:** 2022-03-11

**Authors:** Vanessa Mancebo-Campos, María Desamparados Salvador, Giuseppe Fregapane

**Affiliations:** Department of Analytical Chemistry and Food Technology, Faculty of Chemistry, University of Castilla-La Mancha, 13071 Ciudad Real, Spain; amparo.salvador@uclm.es (M.D.S.); giuseppe.fregapane@uclm.es (G.F.)

**Keywords:** extra virgin olive oil, shelf-life, olive oil storage, accelerated shelf-life test, olive oil stability, antioxidants, prediction model, oxidation kinetics

## Abstract

The development of effective shelf-life prediction models is extremely important for the olive oil industry. This research is the continuation of a previous accelerated shelf-life test at mild temperature (40–60 °C), applied in this case to evaluate the oxidation effect of temperature on minor components (phenols, tocopherol, pigments) to properly complete a shelf-life predictive model. The kinetic behaviour of phenolic compounds, α-tocopherol and pigments during storage of different virgin olive oil samples at different temperatures (25–60 °C) is reported. Hydroxytyrosol, tyrosol and α-tocopherol fitted to pseudo-zero-order kinetics, whereas secoiridoid derivatives of hydroxytyrosol and tyrosol, o-diphenols and total phenols apparently followed pseudo-first-order kinetics. The temperature-dependent kinetic of phenolic compounds and α-tocopherol were well described by the linear Arrhenius model. The apparent activation energy was calculated. Principal component analysis was used to transform the considered compositional and degradation variables into fewer uncorrelated principal components resulting in 4: “no oxidizable substrate”, “initial oxidation state and conditions”, “free simple phenols”, and “degradation rates”. In addition, multivariate linear regression was used to yield several modelling equations for shelf-life prediction, considering initial composition and experimental variables easily determined in accelerated storage.

## 1. Introduction

The shelf-life of food products is strongly related to sensory quality stability and microbiological spoilage. In the case of virgin olive oils (VOO), it is very inhospitable for microbes, so they do not suffer microbiologic degradation. VOO oxidation is the main cause of the reduction in its extraordinary quality as vegetal oil. From its extraction until its consumption, it is highly dependent on factors including the techniques of oil extraction [1,2], exposure of pastes and oils in mills [3], and storage conditions of the final product [4]. The main agents affecting the oxidation degree are exposure to oxygen, light and temperature [5], although the materials in contact with oil also have a role in oxidation development [6].

Shelf-life was defined as the length of time under normal storage conditions within which no off-flavours or defects are developed, and quality parameters are within acceptable limits for this commercial category [7]. Extra virgin olive oil is appreciated for its nutritional value from monounsaturated fatty acids and natural antioxidants [8], as well as for its pleasant organoleptic profile. In 2011, the EFSA (European Food Safety Authority) approved the health claim “olive oil polyphenols (standardised by the content of hydroxytyrosol and its derivatives) protect LDL particles from oxidative damage” [9] and may be used for olive oil that contains at least 5 mg of hydroxytyrosol and its derivatives per 20 g of olive oil [10]. Thus, shelf-life should also be related to the persistence of these compounds.

Off-flavours (defects) and values of standardised parameters that provide olive oil with its commercial quality degree [11] are mainly related to the increase of primary and secondary oxidation products formed from fatty acids. However, since oxidation reactions have a negative impact on phenolic compounds, tocopherols, pigments and other olive oil components with sensory and health attributes, it seems to be meaningful to contemplate the disappearance of these added-value compounds during storage to the suitable establishment of VOO commercial shelf-life.

It has been reported that extrapolation from accelerated shelf-life tests (ASLT) at high temperatures, as Rancimat or OSI values, led to either underprediction or overprediction of the real shelf-life of sunflower and olive oil due to the drastic conditions [12]. That reason encouraged many authors to develop ASLT at mild temperatures (lower than 60 °C) [13] in parallel to shelf-life tests at ambient conditions [14], since shelf-life prediction models are best developed based on results from both real-time and accelerated storage conditions.

Several trials to predict the shelf-life of olive oils from kinetic or empirical models have been developed in this time [15,16]. Zanoni et al. [17] reported for the first time a phenomenological model to predict the stability of VOO based on combined initial composition indices. The selected degradation parameters could be predicted by means of acidity, oleic acid and bitter taste, but no longer the initial state, since after storage, degradation parameters may change as a result of lipid oxidation, and models based on constant indices are not able to predict a new degradation extent.

This research group, in a previous paper [18], described an ASLT carried out in the dark at mild temperatures (40, 50, and 60 °C), where the autoxidation kinetic behaviour of the main oxidation indices (PV, K232, and K270) and the oxidising substrate (unsaturated fatty acids (UFA)) were reported for the first time. K232 showed high linearity in the early stages of oxidation and presented an excellent correlation with the loss of UFA. Thus, K232 was selected as the best-normalised oxidation index for potential shelf-life estimation of VOO, defined as TRUL (Time to Reach the K232 legal Upper Limit of 2.50) at a mild temperature (≤60 °C):TRUL = a ∙T^b^
(1)

The current work is the advancement of that predictive study and focuses now on the contribution of the initial antioxidants and fatty acids contents to lipid oxidation rates, describing as well minor components degradation rates, to estimate VOO shelf-life from its initial composition and its oxidation progress.

## 2. Materials and Methods

### 2.1. Virgin Olive Oil (VOO) Samples

Five virgin olive oils of the Cornicabra variety (III-VII) were kindly supplied for industrial oil mills located in Toledo and Ciudad Real (Castilla-La Mancha, Spain). Two more virgin olive oils (I-II) were obtained from Cornicabra olives using the Abencor system (Comercial Abengoa, S.A., Sevilla, Spain) to produce oils with a higher concentration of phenolic compounds. All samples were filtered with anhydrous Na_2_SO_4_ and stored in darkness at 8 °C using amber glass bottles without headspace until analysis.

### 2.2. Oxidation Experiments

Aliquots of 40 mL (36.6 g) of each VOO were stored in darkness in 125 mL open amber glass bottles (i.d.: 4.2 cm; surface area exposed to the air: 13.85 cm^2^) at 25, 40, 50 and 60 °C during 93, 41, 34 and 19 weeks, respectively. One bottle was taken from the incubator for analysis at scheduled times. Two individual batches of samples for each temperature condition studied were used.

### 2.3. Analytical Determinations

All reagents used were of analytical, HPLC or spectroscopic grade and were supplied by Merck (Darmstadt, Germany). All experiments and analytical determinations were carried out at least in duplicate.

#### 2.3.1. Peroxide Value (PV), K232 and K270

PV expressed as milliequivalents of active oxygen per kilogram of oil (meq O_2_/kg), and K232 and K270 extinction coefficients calculated from absorption at 232 and 270 nm were measured following the analytical methods described in European Regulation [19].

#### 2.3.2. Phenolic Compounds

250 µL of a solution of the internal standard (syringic acid in methanol, 15 mg/L) was added to a sample of virgin olive oil (2.5 g), and the solvent was evaporated with a rotary evaporator at 35 °C under vacuum. The oil was then dissolved in 6 mL of n-hexane, and a diol-bonded phase cartridge (Supelco Co., Bellefonte, PA, USA) was used to extract the phenolic fraction. The cartridge was conditioned with methanol (6 mL) and n-hexane (6 mL), the oil solution was then applied, and the SPE column was washed with n-hexane (2 × 3 mL) and with n-hexane/ethyl acetate (85:15, *v*/*v*; 4 mL). Finally, the phenolic compounds were eluted with methanol (15 mL), and the solvent was removed with a rotary evaporator at 30 °C under vacuum until dryness. The phenolic residue was dissolved in methanol/water (1:1 *v*/*v*; 250 µL).

HPLC analysis was performed using an Agilent Technologies 1100 series system equipped with an automatic injector, a column oven and a diode array UV detector. A Spherisorb S3 ODS2 column (250 × 4.6 id mm, 5 µm particle size) (Waters Co., Milford, MA, USA) was used, maintained at 30 °C. The injection volume was 20 µL, and the flow rate was 1.0 mL/min. Mobile phase was a mixture of water/acetic acid (95:5 *v*/*v*) (solvent A), methanol (B) and acetonitrile (C): from 95% (A) −2.5% (B) −2.5% (C) to 34% (A) −33% (B) −33% (C) in 50 min. Phenolic compounds were quantified at 280 nm using syringic acid as internal standard and the response factors determined by Mateos et al. [20].

#### 2.3.3. Tocopherols

Were evaluated following AOCS Method Ce 8–89. A solution of oil in n-hexane was analysed on an Agilent Technologies HPLC (1100 series) on a silica gel Lichrosorb Si-60 column (particle size 5 µm, 250 mm × 4.6 mm i.d.; Sugerlabor, Madrid, Spain), which was eluted with n-hexane/2-propanol (98.5:1.5) at a flow rate of 1 mL/min. A fluorescence detector (Thermo-Finnigan FL3000, Waltham, MA, USA) was used with excitation and emission wavelength set at 290 and 330 nm, respectively.

#### 2.3.4. Fatty Acid Composition

To determine fatty acid composition, the method described in Mancebo-Campos et al. [21] was used. The loss in the unsaturated fatty acids due to oxidation was quantified on the basis of the ratio between each fatty acid and the palmitic acid peak areas since saturated fatty acids are not altered by autoxidation [22].

#### 2.3.5. Chlorophyll and Carotenoid Compounds

These compounds (mg/kg) were determined at 472 and 670 nm in cyclohexane using specific extinction values by the method of Minguez-Mosquera et al. [23].

#### 2.3.6. Oxidative Stability

This was evaluated by the Rancimat method [24]. Stability was expressed as the induction time (hours) measured with the Rancimat 679 apparatus (Metrohm, Switzerland) at 100 °C and 10 L/h airflow.

### 2.4. Statistical Analysis and Treatment of Experimental Data

The experimental data set consisted of two individual batches of samples for each of the temperature conditions studied. Moreover, duplicate measurements were carried out for each sample taken at scheduled time intervals. Linear and nonlinear regression analyses were performed by using Microsoft Office Excel 2007 for Windows (Microsoft Corporation, Redmond, WA, USA), and the best-fitted equations were selected on the basis of statistical parameters of the studied regression (R, *p*). Statistical analysis (PCA and MLR) was performed with the SPSS 14 statistical software (SPSS Inc., Chicago, IL, USA). One-way ANOVA was carried out using the Duncan test. Means were considered statistically different at *p* < 0.05. Antioxidant degradation rates were calculated from the slopes of the respective concentration vs. time experimental curves.

The effect of temperature on the rates of reaction was evaluated by means of the Arrhenius Equation [25]:(2)$$Ln\;\;k=Ln\;\;A\;-\;\;\frac{\mathrm{Ea}}{\mathrm{RT}}$$
where k is the reaction rate constant, R is the molar gas constant (8.31 J K^−1^ mol^−1^), T is the absolute temperature (K), Ea is the activation energy (J mol^−1^), and A is the pre-exponential factor. Since some level of curvature in the *Ln* k vs. 1/T plot is possible in some cases, it is feasible to use a modified equation [26,27]:(3)$$k={\mathrm{AT}}^{n}\;e^{-\mathrm{Ea}/\mathrm{RT}}$$
where A, n and Ea are parameters determined using nonlinear fitting (0 < n < 1).

## 3. Results and Discussion

### 3.1. Initial Characteristics of Virgin Olive Oils

As reported in Table 1, all the VOO samples met the European Union requirements for the “extra” virgin category [28]. The initial PV (≤6.5), K232 (≤1.93) and K270 (≤0.16) showed that oxidation level was low and very similar in the 7 virgin olive oils studied. Lipid matrices were very similar since all samples came from monovarietal VOO grown in the same area (Cornicabra cultivar in Castilla-La Mancha, Spain). Nevertheless, there were some expected low but statistically significant differences in the unsaturated fatty acid (UFAs) contents of samples, especially in linoleic acid (C18:2). On the contrary, and according to the purpose of this study, there were notable differences in the types and concentrations of natural antioxidants, phenolic compounds and tocopherols.

Table 1 depicts in detail the phenolic profile of VOO samples and their content of α-tocopherol. It can be seen that samples II and III had the highest total phenol content (3.88 mmol/kg), followed by sample I (3.70 mmol/kg). However, sample II had a higher content of dialdehydic forms of oleuropein and ligstroside aglycons and lower of the aldehydic forms, whereas it occurs inversely in sample III.

The lowest total phenol content was that of samples VII (1.08 mmol/kg) and VI (1.35 mmol/kg); however, the latter presented the highest content of free tyrosol (0.54 mmol/kg) and hydroxytyrosol (0.26 mmol/kg) and the lowest one of complex phenolic compounds (0.55 mmol/kg). As many authors have reported [20,29,30], o-diphenols are mainly responsible for the oxidative stability of virgin olive oils, and that is why these samples, which differ in the o-diphenol content, were chosen. Samples IV and V were very similar with respect to the o-diphenols and total phenol content; however, they presented very different complex/simple phenol ratios. Sample I had the highest α-tocopherol content (0.55 mmol/kg) and sample VI the lowest one (0.33 mmol/kg).

### 3.2. Kinetic Behaviour of Phenolic Compounds and Pigments Degradation

Figure 1 shows the evolution of hydroxytyrosol content in all samples at 25–40 °C. For a better comparison of differences between samples, data on the ordinate axis were expressed as the decrease in the hydroxytyrosol content with respect to the initial content. As mentioned in previous works of this research group [21,31], at 25 °C, the linear increase of hydroxytyrosol was general, mainly due to the non-oxidative hydrolysis of their secoiridoid derivatives [32]. However, sample VI suffered a reduction in this compound, maybe owing to both the low content of hydroxytyrosol secoiridoids in this sample and the high one of the free hydroxytyrosol, as also reported in Mancebo-Campos et al. [33], where purified olive oil samples with less than 0.50 mmol/kg of added hydroxytyrosol secoiridoids behaved the same.

As the temperature increased, differences between samples concerning their phenolic composition became more evident because of the different rates of hydrolysis of the secoiridoid derivatives, the thermal decomposition and its action as antioxidants. At 40 °C, hydroxytyrosol content increased only in sample II and was maintained practically constant in samples III and IV during the whole experimental period (41 weeks). For the rest of the samples, an initial stage was observed where this simple phenol slightly increased or remained constant, to diminish afterwards at different rates until reaching a final stationary phase. At 50 and 60 °C, the rate of decomposition seems to be higher than that of formation, since the content of hydroxytyrosol falls from the very first in all samples. The initial “steady state”, when present, was shorter at 60 °C. Samples IV, V, VI and VII reached the final stationary state at 50 and 60 °C when hydroxytyrosol had practically disappeared; however, this was not the case of samples I, II and III, which preserved between 20 and 50% of the initial content at the end of the storage period. The faster decrease of hydroxytyrosol at all temperatures was in sample VI, that with lowest ratios Sec. Htyr/Free Htyr (1.21) and Complex/Simple Phenols (0.69), whereas the slowest rates of hydroxytyrosol were those of samples II and IV, which had the highest ratios Sec. Htyr/Free Htyr (21.1 and 31) and Complex/Simple Phenols (18.4 and 25.3). One of the lowest rates was also in sample III, which presented the highest o-diphenol content and highest ratio o-diphenols/total phenols (Table 1). In samples I, II and III, it is feasible a reasonable yielding of hydroxytyrosol from complex phenols hydrolysis.

The content of tyrosol followed a general growing trend at all temperatures studied (Appendix A), meaning that the rate at which this compound decomposed was always lower than that of the hydrolysis of its secoiridoid derivatives. One exception to this trend was sample VII, with the lowest content of tyrosol secoiridoids, in which the hydrolysis of the complex was not enough to compensate for the tyrosol lost either by thermal decomposition or as a consequence of its antioxidant action. The same happened to sample VI from 40 to 60 °C, as shown.

A general decrease of hydroxytyrosol secoiridoids (sum of the dialdehydic and aldehydic forms of oleuropein aglycon) and of o-diphenols (also including free hydroxytyrosol) was common to all samples (Figure 2). In the final state, the content of these secoiridoids maintained practically invariable or dropped very slowly, coinciding with the stabilisation of peroxide values and K232 reported for the same samples in previous work [18,34].

The tendency of tyrosol secoiridoids (sum of the dialdehydic and aldehydic forms of ligstroside aglycon) content was similar to that of hydroxytyrosol derivatives (Appendix A), but the rate of decrease of the former and the percentage of final losses were lower, so the stability of the tyrosol secoiridoid compounds appeared to be greater than that of hydroxytyrosol derivatives, according to previous works [21,33,35].

The experimental data confirm that the o-diphenol group comprised in the molecule of hydroxytyrosol and its derivatives makes them more active antioxidants in the used conditions but possibly also more susceptible to oxidation and thermal decomposition than tyrosol secoiridoids as previously reported [18,21,33], as demonstrated by the lower slope of the degradation kinetics, especially at higher storage temperatures (Appendix A).

The analysis of time (t) and concentration (C_t_) data indicated that evolution of hydroxytyrosol and tyrosol fitted to pseudo-zero-order kinetics (C_t_ = C_0_ + *k*t), whereas the secoiridoids of hydroxytyrosol and tyrosol, o-diphenols and total phenols apparently followed pseudo-first-order kinetics (Ln(C_t_/C_0_) = *k*t), in accordance with Lavelli et al. [36], Gómez-Alonso et al. [31], Mancebo-Campos et al. [18] and Krichene et al. [35]. Kinetic parameters are detailed in Appendix A. Degradation rate constant (*k*) values obtained in this study for secoiridoids of tyrosol and hydroxytyrosol are around 100 times higher than those obtained for Lavelli et al. [36] at 40 °C, since they used closed bottles but were in agreement with those of Krichene et al. [35] in open bottles at 25 °C and 50 °C in samples with similar initial phenolic content.

The content of α-tocopherol followed a similar pattern in all samples, falling roughly linearly and faster with temperature (Figure 3). At 25 °C and 40 °C, the rate of reduction seemed to be slightly lower at the beginning of storage, seemed to remain nearly constant during more than 40 weeks of storage at 25 °C, and began to go down when o-diphenols depletion had almost concluded. This is in accordance with that stated in [21]: α-tocopherol showed antioxidant effect at advanced oxidation stages when free peroxyl radicals reach a certain value and are not trapped for phenolic antioxidants. However, at 50 and 60 °C, α-tocopherol content dropped rapidly from the beginning and stabilised in all samples when the depletion was significant. The total decrease of this compound ranged between 12–28% at 25 °C, 26–56% at 40 °C, 79–99% at 50 °C and 90–100% at 60 °C.

Some authors have reported the high susceptibility of this molecule to oxidise to α-tocopherol quinones at high temperatures [37], a fact that could explain the faster decrease at 50 and 60 °C. Tocopherol at temperatures lower than 50 °C is protected for polar phenols acting as antioxidants in the first oxidation stages.

Excluding the final stationary phases at 50 and 60 °C, the evolution of α-tocopherol was attempted to correlate to a pseudo-zero-order reaction (according to [35]). Experimental rate constants (*k*) are listed in Appendix A.

Carotenoid pigments are well known as photooxidation protectors by quenching singlet oxygen and acting as light filters [38]. The activity during autoxidation is complex due to their susceptibility to oxidation [39] because of the presence of a conjugated double bond system and hydroxyl groups in its molecule. This fact could explain the higher rate of reduction with temperature displayed in Appendix A, in accordance with Hrncirik, et al. [40]. The effect of carotenoids in VOO under conditions of autoxidation could be even negative due to their oxidation products, which may possibly react with the lipid substrate and thus accelerate oxidation [41].

The reduction of chlorophylls content was much lower than that of carotenoids, according to Ceballos et al. [42] and Hrncirik et al. [40]. Their effect as photosensitisers should be non-existent in this study since the oils were stored in darkness. However, it may be considered the antioxidant activity of these compounds in darkness due to de possible donation of a hydrogen radical to break free-radical chain reactions [43,44]. The decrease could also be because in olive oil, chlorophyll pigments degrade to form pyropheophytins; this reaction begins soon after the oil is extracted. The pigments break down due to a process that involves the decarbomethoxylation of chlorophyll and pheophytins to form pyropheophytins [45].

The degradation of pigments results in visual changes in the colour of VOO samples only perceptible at high temperatures; thus, colour changes at 25 °C would not be a determining factor for VOO shelf-life.

### 3.3. Feasibility of the Arrhenius Equation

As expected, and shown in Appendix A, an increase in storage temperature increased the degradation rate (*k*) of phenolic compounds and α-tocopherol, particularly at 50 and 60 °C.

Regression analyses from the Ln *k* vs. 1/T plots indicated that the temperature dependence of phenolic compounds and α-tocopherol degradation rates were well described by the linear Arrhenius model between 25 and 60 °C, but regression for tyrosol kinetic presented the worst correlation factors (Table 2).

The apparent activation energy (Ea) was calculated from the slopes of the lines fitted to the Ln *k* plotted as a function of the inverse of absolute temperature (Table 2 and Table 3). Values of Ea were not significantly different between samples; thus, similar energy was necessary to initiate the degradation of phenolic compounds and tocopherol despite the initial composition. These Ea values for total phenols degradation were nearly twice those reported for Campanella et al. [46] (37.4−39.1 kJ mol^−1^), with olive oil forced to oxidation at temperatures higher than 98 °C. Ea values for α-tocopherol were in all samples higher than those of phenolic compounds, showing again the higher stability of this compound in olive oil when phenols are present.

The Ea obtained previously for oxidation reactions (65 kJ/mol for primary oxidation products and of about 77 kJ/mol for secondary oxidation products) [18] of the same olive oil samples indicated no marked relationship with oxidative stability or TRUL.

Accordingly, neither Ea for antioxidants decay showed a relationship with compositional indicators (MUFA/PUFA, or antioxidant contents). This suggests that Ea should not be used as a single parameter to compare the rate of lipid oxidation or the oxidative stability of olive oil or other lipid systems [13].

The other main kinetic parameter affecting reaction rate is A (pre-exponential or frequency factor), calculated from the intercepts of the lines fitted to the Ln *k* plotted as a function of the inverse of absolute temperature. Minor changes in Ea leads to major changes in A values. In fact, plots of Ea vs. A leads to exponential trend (0.972 < R^2^ < 0.998). This means a higher contribution for A than for Ea to the rate of lipid oxidation and the rate of antioxidant depletion of the oils studied, according to Uri [47] and Cho [48]. In fact, there were higher and significant differences in A values between samples for all phenolic compounds and α-tocopherol (Table 2 and Table 3). A, being an indicator of the frequency of molecules collisions, could be thought to have a relationship with initial concentration; however, no meaningful correlation was found.

The obtained values of Ea and *k* cannot provide a mechanistic interpretation of any reaction since oxidation is a complex reaction, but they can be used as descriptive tools of the temperature dependence of reaction [49].

### 3.4. Shelf-Life of Virgin Olive Oil Related to Its Initial Composition

From the study of oxidation parameters and fatty acids degradation rates between 25 and 60 °C, this research group proposed the TRUL (Time to Reach the Upper legal Limit) parameter, related to K232, as a value for predicting oxidative stability at 25 °C, from ASLT at a temperature lower than 60 °C [18]. As a result, the predicted TRUL at 25 °C when applying the proposed model to accelerated storage temperatures (40, 50 and 60 °C) was very close to the experimental TRUL at the same temperature.

The high MUFAs/PUFAs ratio, which is typical of olive oil, is one of the main reasons for the higher stability of olive oil with respect to other edible oils [50,51]. In the studied samples, this ratio ranged from 13.5 to 19.4. The higher levels of the ratio should show the greater oxidative stability of the olive oils, but in this previous study [18], no direct correlation was found between this ratio and the rate of oxidation at temperatures below 60 °C, or with shelf-life determined by the oxidation parameters.

Concerning phenolic compounds and their relationship with oxidative stability, the high correlation of these compounds has been widely reported, mainly o-diphenols with higher antioxidant capacity, with the induction period measured by Rancimat [30]. Nevertheless, this correlation is not simple and evident with the rate of oxidation at temperatures below 60 °C and neither with shelf-life determined by the oxidation parameters [18,31]. This could be because the content of phenolic compounds in VOOs exceeds the point to which the concentration of antioxidants is still relevant as oxidation rate is concerned.

In the current work, further investigation was carried out concerning the subjacent relationship between oxidative stability at room temperature of virgin olive oil and its initial composition, describing minor components degradation rates, to estimate VOO shelf-life from its initial composition and its oxidation progress.

The great number of variables related to initial composition and oxidative degradation were treated to transform into fewer uncorrelated principal components by Principal component analysis (PCA). Then, multivariate linear regression (MLR) was performed to describe the relationship between shelf-life and meaningful variables or the obtained principal components.

Variables used to indicate the extent or progress of oxidation were those related to the increase of primary and secondary oxidation products and the decrease of antioxidant compounds or unsaturated fatty acids. These are detailed in Table 4, as well as variables regarding the initial lipidic or antioxidant composition.

A PCA (PCA1) was performed, including the variables related to the initial physical-chemical state of samples (16 variables, see Table 4) and temperature, separately at 25, 40, 50 and 60 °C (*N* = 28 samples), and with joined data for the 4 temperatures (*N* = 112 samples). For *N* = 112, the KMO factor (Kayser-Meyen-Olkin) had a value of 0.773, indicating an acceptable fit of the sample to this analysis. The Bartlett test had 0 significance, so there was significant correlation between variables. Three factors (principal components) were generated in the PCA, explaining 79.11% of the variance. Significant eigenvalues were PC1 = 9.035, PC2 = 2.932, PC3 = 1.481.

From the rotated component matrix (Table 5), the highest and positive loadings in PC1 were those of initial antioxidants contents (chlorophyll, carotenoids, hydroxytyrosol secoiridoids, tyrosol secoiridoids, o-diphenols, tyrosol + secoiridoids, total phenols) and directly related measures (Rancimat OS). The highest and negative loadings were those of UFAs and PUFAs contents (oxidizable substrate). Thus, PC1 could be considered as the factor “non-oxidizable substrate” and explained the 54.15% of variance.

The highest and positive loadings in PC2 were those of temperature, initial PV, K232 and K270. PC2 could be the factor “initial oxidation state and conditions” and explained the 17.25% of the variance.

The highest and positive loadings in PC3 were those of initial hydroxytyrosol and tyrosol contents, so PC3 could be the factor “free simple phenols”, explaining 8.72% of the variance.

A new PCA (PCA2) was used to reduce variables based on rates of increase or decrease of oxidation indices, fatty acids and antioxidants (10 variables, see Table 4), separately at 25, 40, 50 and 60 °C (*N* = 28 samples), and with joined data for the 4 temperatures (*N* = 112 samples). For *N* = 112, the KMO factor had a value of 0.885, indicating a good fit of the sample to this analysis. The Bartlett test had 0 significance, so there was significant correlation between variables. Applying the Kaiser rule (eigenvalues < 1), only one factor (principal component) was generated in the PCA, explaining 87.2% of the variance. Significant eigenvalue was PC4 = 9.594. In this case, loadings were negative for temperature and rates of PV, K232, K270, and were positive for rates of antioxidant and PUFAs reduction (Appendix A), so this factor could mean “degradation rates”.

Several attempts of multivariate linear regression (MLR) were done in order to explain the relationship among shelf-life, initial composition and/or oxidation progress. For sequential variable selection, a stepwise regression was carried out. An analysis of variance (ANOVA) procedure was used to determine the significance of the model. The results of the MLR model respond to an equation like:y = h_0_ + b_1 × 1_ + b_2 × 2_ + b_3 × 3_ +……. b_n_X_n_
(4)
where y is dependent variable, X_n_ are the independent variables, and b_n_ corresponded to coefficient correlation of X_n_. A negative b_n_ means a preventive or negative effect of the corresponding variable on y. On the other hand, positive values demonstrate some correspondence with the selected dependent variable. h_0_ in the final equation correspond to the constant. To select the proper model equations, diagnosis of collinearity, homoedasticity and normal residues distribution were tested.

The parameters suggested as dependent variables were those related to VOO quality:-tPV_25: time needed at 25 °C to reach the upper legal limit for peroxide value in extra virgin olive oils (20 meq/kg);-tK232_25: time needed at 25 °C to reach the upper legal limit for K232 in extra virgin olive oils (2.50);-tK270_25: time needed at 25 °C to reach the upper legal limit for K270 in extra virgin olive oils (0.22).

Considering also that the health claim “olive oil polyphenols (standardised by the content of hydroxytyrosol and its derivatives) protect LDL particles from oxidative damage” may be used for olive oil that contains at least 5 mg of hydroxytyrosol and its derivatives per 20 g of olive oil [9,10], other investigated dependent variables were:-kodPh: rate of o-diphenols (hydroxytyrosol and derivatives) decrease at 25 °C;-ksecHTyr: rate of hydroxytyrosol secoiridoid derivatives decrease at 25 °C.

This calculation and the known initial content of o-diphenols or hydroxytyrosol secoiridoid derivatives could predict the time to which the health claim could be maintained.

First, the calculated factor for principal components PC1 “non-oxidizable substrate: fatty acid and antioxidant content”, PC2 “initial oxidation state and conditions”, PC3 “simple phenols” and PC4 “degradation rates”, were considered as independent variables. The stepwise MLR gave the first 4 models shown in Table 6. For both, time to reach PV = 20 and time to reach K232 = 2.50, there were positive and highest coefficients for “degradation rates”; positive and lowest coefficient for “non-oxidizable substrate”; negative and highest for “initial oxidation state and conditions”, and negative and lowest for “free simple phenols content”. For the calculation of dependent variables degradation rates of o-diphenols and hydroxytyrosol derivatives, there were positive coefficients for “degradation rates” and “non-oxidizable substrate”. Although high correlation factors were obtained for kodPh_25 and ksecHTyr_25 (0.937 and 0.969, respectively), these equations had some value for explaining the behaviour of the experimental variables, but little predictive value due to the complexity for factor experimental calculation.

In a second step, there were considered as independent variables all those related to the initial state, besides those related to oxidation development at accelerated temperatures studied (40, 50 and 60 °C) (see Table 4). Throughout the stepwise MLR, the software sometimes introduced just one independent variable that did not significatively increase R^2^ but supposed to perform an accelerated oxidation test at a different temperature. In this case, this variable was removed from the model to simplify the experimental predictive work. Moreover, independent variables with a VIF (Variance Inflation Factor) value higher than 5 in the collinearity test were not included in the model. The last 4 models shown in Table 6 are the selected model equations from measurable experimental variables.

The models explained a high percentage of the variance (88.1–91.5%) with few independent variables automatically considered among all available, which simplifies the experimental calculation of the dependent variables.

On the basis of the experimental results observed in a previous study [18], it is feasible to perform an accelerated stability test (ASLT) at a temperature below 60 °C to estimate real shelf-life based on normalised parameters, using TRUL (time to reach K232 = 2.50 at 25 **°**C, or tK232_25) and a mean empirical value for factor “b = **−**3.72 ± 0.22”:TRUL = tK232_25 = a T^b^
(5)
where T is the temperature at ambient storage (i.e., 25 °C)

The factor “a” can be experimentally calculated from the selected ASLT:a = tK232__exp_/T_exp_ ^(−3.72 ± 0.22)^
(6)
where tK232__exp_ is the time to reach K232 = 2.5 at the experimental temperature (T_exp_) of the accelerated test (i.e., 40 **°**C);

Considering also the model obtained by MLR, that take into account initial composition and degradation rates, the tK232_25 can be also predicted by means of the following model equation:tK232_25 = 6.23 + 7.63 tK232_40 + 490.05 koDIPH_40 (R^2^ = 0.904)(7)

The moment the concentration of hydroxytyrosol derivatives would fall off below 5 mg/20 g of olive oil can also be estimated, using the initial content of these compounds and by prediction of the degradation rate of o-diphenols at room temperature from accelerated storage at 50 °C, by means of the model equation:kodPh_25 = −0.008 − 0.005 kPV_50 + 0.001PV − 0.001 tK_232__50 (R^2^ = 0.901)(8)

Figure 4 shows the correlation between real experimental values and those calculated with model equations.

## 4. Conclusions

In order to elucidate a model that explains relationships between virgin olive oil oxidation parameters, initial composition and progressive degradation of major and minor compounds, this work and the previous one of this research group [18] widely described the initial characteristics and oxidation progress at 25, 40, 50 and 60 °C.

A simple mathematical equation is feasible to be used to predict the time to reach the quality index K232 = 2.50 at room temperature, from the same parameter calculated by a shorter but useful ASLT at less than 60 °C.

However, it seems clear that the composition of the VOO, the initial oxidation state and the progressive degradation of the antioxidants should contribute to the rate of increase in the oxidation indices, so this work investigates a model that relates them mathematically.

The presented models allow us to calculate, for example, the time to reach K232 = 2.50 at room temperature from an ASLT at 40 °C, also considering the rate of degradation of o-diphenols at 40 °C

Moreover, knowing the initial content of hydroxytyrosol derivatives, the moment the concentration would fall off below 5 mg/20 g of olive oil can also be estimated by one of the proposed models that relate the degradation rate of o-diphenols at room temperature with the measure of PV and K232 during accelerated storage at 50 °C.

## Figures and Tables

**Figure 1 antioxidants-11-00539-f001:**
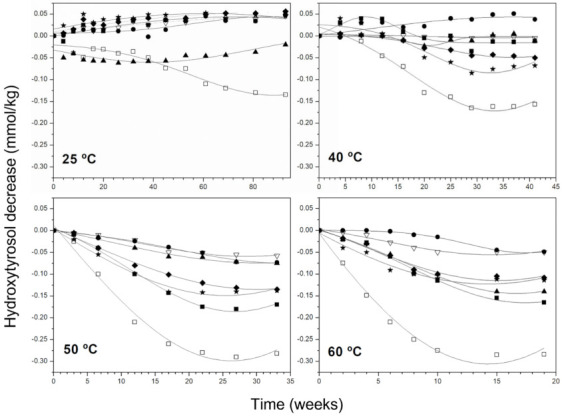
Decrease of hydroxytyrosol content in VOO samples at 25, 40, 50 and 60 °C. Samples: ■, I; •, II; ▲, III; ∇, IV; ◆, V; □, VI; ★, VII.

**Figure 2 antioxidants-11-00539-f002:**
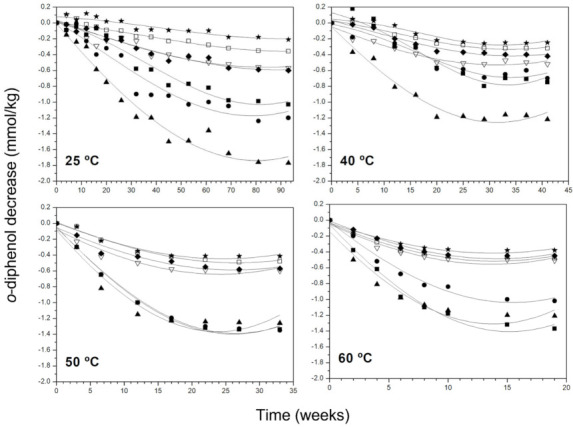
Decrease of o-diphenol content in VOO samples at 25, 40, 50 and 60 °C. Samples:■, I; •, II; ▲, III; ∇, IV; ◆, V; □, VI; ★, VII.

**Figure 3 antioxidants-11-00539-f003:**
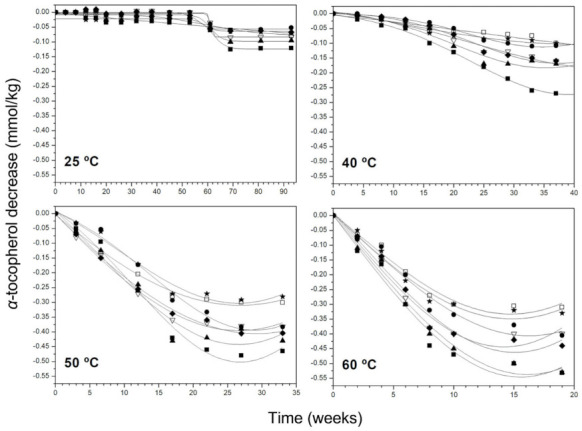
Decrease of α-tocopherol content in VOO samples at 25, 40, 50 and 60 °C. Samples: ■, I; •, II; ▲, III; ∇, IV; ◆, V; □, VI; ★, VII.

**Figure 4 antioxidants-11-00539-f004:**
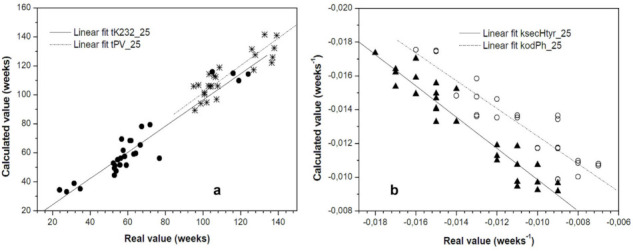
Real and calculated values from predictive equations. •, tK232_25; ж, tPV_25; ▲; ksecHtyr_25; ○; kodPh_25. (**a**) real experimental values; (**b**) values calculated with model equations.

**Table 1 antioxidants-11-00539-t001:** Initial Composition and Quality Indexes of Olive Oil Samples.

	VOO Samples						
	I	II	III	IV	V	VI	VII
PV (meq O_2_/kg)	5.5 ± 0.04 ^d^	5.5 ± 0.01 ^d^	3.1 ± 0.01 ^b^	3.7 ± 0.01 ^c^	2.9 ± 0.01 ^a^	6.5 ± 0.01 ^f^	5.8 ± 0.00 ^e^
*K* _232_	1.93 ± 0.03 ^f^	1.87 ± 0.04 ^e^	1.73 ± 0.01 ^d^	1.61 ± 0.01 ^a,b^	1.56 ± 0.01 ^a^	1.67 ± 0.03 ^b,c^	1.72 ± 0.01 ^c,d^
*K* _270_	0.16 ± 0.01 ^e^	0.16 ± 0.01 ^e^	0.13 ± 0.00 ^c,d^	0.12 ± 0.01 ^b,c^	0.10 ± 0.01 ^a^	0.12 ± 0.00 ^b^	0.14 ± 0.00 ^c,d^
C18:1 (%)	79.00 ± 0.04 ^a^	81.52 ± 0.01 ^g^	81.02 ± 0.01 ^f^	79.37 ± 0.03 ^b^	80.92 ± 0.01 ^e^	80.07 ± 0.01 ^c^	80.75 ± 0.01 ^d^
C18:2 (%)	5.09 ± 0.01 ^d^	3.61 ± 0.01 ^a^	3.87 ± 0.01 ^b^	5.29 ± 0.01 ^e^	5.32 ± 0.01 ^f^	4.94 ± 0.01 ^c^	5.31 ± 0.01 ^e,f^
C18:3 (%)	0.71 ± 0.01 ^e^	0.59 ± 0.01 ^c^	0.65 ± 0.01 ^d^	0.58 ± 0.01 ^b^	0.58 ± 0.01 ^b^	0.54 ± 0.01 ^a^	0.57 ± 0.01 ^b^
Chlorophyll ^(1)^	11.40 ± 0.10 ^e^	42.97 ± 0.09 ^g^	19.57 ± 0.04 ^f^	3.83 ± 0.07 ^c^	2.20 ± 0.04 ^a^	6.79 ± 0.17 ^d^	2.58 ± 0.02 ^b^
Carotenoids ^(1)^	6.32 ± 0.01 ^e^	15.27 ± 0.01 ^g^	12.89 ± 0.01 ^f^	4.12 ± 0.00 ^c^	2.71 ± 0.00 ^a^	4.49 ± 0.01 ^d^	3.09 ± 0.01 ^b^
Htyr ^(2)^ (3,4-DHPEA)	0.16 ± 0.00 ^d^	0.10 ± 0.00 ^c^	0.19 ± 0.02 ^e^	0.03 ± 0.00 ^a^	0.06 ± 0.00 ^b^	0.27 ± 0.01 ^f^	0.07 ± 0.00 ^b^
3,4-DHPEA-EDA	1.48 ± 0.00 ^d^	1.60 ± 0.21 ^d^	0.89 ± 0.03 ^c^	0.30 ± 0.03 ^b^	0.34 ± 0.01 ^b^	0.05 ± 0.01 ^a^	0.19 ± 0.00 ^a,b^
3,4-DHPEA-EA	0.40 ± 0.00 ^c^	0.47 ± 0.04 ^d^	1.38 ± 0.05 ^e^	0.38 ± 0.01 ^c^	0.31 ± 0.01 ^b^	0.27 ± 0.00 ^a,b^	0.25 ± 0.01 ^a^
Sec. Htyr ^(2)^	1.88 ± 0.00 ^d^	2.07 ± 0.25 ^d,e^	2.27 ± 0.02 ^e^	0.68 ± 0.05 ^c^	0.64 ± 0.02 ^b,c^	0.32 ± 0.01 ^a^	0.43 ± 0.00 ^a,b^
Tyr ^(2)^ (p-HPEA)	0.12 ± 0.00 ^c^	0.10 ± 0.01 ^b^	0.12 ± 0.01 ^c^	0.04 ± 0.00 ^a^	0.06 ± 0.01 ^a^	0.54 ± 0.01 ^d^	0.09 ± 0.01 ^b^
p-HPEA-EDA	1.29 ± 0.00 ^d^	1.36 ± 0.17 ^d^	0.73 ± 0.03 ^c^	0.47 ± 0.01 ^b^	0.45 ± 0.00 ^b^	0.06 ± 0.01 ^a^	0.33 ± 0.01 ^b^
p-HPEA-EA	0.24 ± 0.00 ^c,d^	0.26 ± 0.03 ^d^	0.58 ± 0.04 ^e^	0.21 ± 0.01 ^b,c^	0.17 ± 0.00 ^a,b^	0.18 ± 0.01 ^a,b^	0.16 ± 0.00 ^a^
Sect. Tyr ^(2)^	1.53 ± 0.00 ^e^	1.61 ± 0.20 ^e^	1.30 ± 0.01 ^d^	0.68 ± 0.02 ^c^	0.63 ± 0.01 ^b,c^	0.24 ± 0.01 ^a^	0.49 ± 0.01 ^b^
o-diphenols ^(2)^	2.05 ± 0.00 ^b^	2.16 ± 0.25 ^b^	2.46 ± 0.00 ^c^	0.71 ± 0.04 ^a^	0.71 ± 0.03 ^a^	0.58 ± 0.01 ^a^	0.51 ± 0.01 ^a^
Total phenols ^(2)^ *	3.70 ± 0.00 ^c^	3.88 ± 0.45 ^c^	3.88 ± 0.00 ^c^	1.41 ± 0.06 ^b^	1.38 ± 0.03 ^b^	1.35 ± 0.02 ^b^	1.08 ± 0.00 ^a^
o-diphenols/Total phenols	0.55 ± 0.03 ^d^	0.56 ± 0.11 ^d^	0.63 ± 0.06 ^e^	0.51 ± 0.07 ^c^	0.51 ± 0.03 ^c^	0.43 ± 0.03 ^a^	0.47 ± 0.02 ^b^
Sec. Htyr/Free Htyr	11.53 ± 0.11 ^d^	21.14 ± 0.12 ^e^	12.23 ± 0.10 ^d^	31.00 ± 0.18 ^f^	10.46 ± 0.12 ^c^	1.21 ± 0.03 ^a^	5.95 ±0.08 ^b^
Complex/Simple Phenols	11.99 ± 0.11 ^c^	18.43 ± 0.13 ^d^	11.55 ± 0.15 ^c^	25.30 ± 0.19 ^e^	11.42 ± 0.12 ^c^	0.69 ±0.02 ^a^	6.02 ± 0.09 ^b^
α-Tocopherol ^(2)^	0.55 ± 0.01 ^e^	0.44 ± 0.01 ^c^	0.53 ± 0.01 ^e^	0.38 ± 0.01 ^b^	0.45 ± 0.01 ^d^	0.33 ± 0.01 ^a^	0.36 ± 0.01 ^b^
α-Tocopherol/o-diphenols	0.27 ± 0.05 ^b^	0.20 ± 0.03 ^a^	0.22 ± 0.02 ^a^	0.54 ± 0.05 ^c^	0.63 ± 0.04 ^e^	0.57 ± 0.04 ^d^	0.71 ± 0.03 ^f^
Stability (h)	133.2 ± 2.5 ^d^	158.0 ± 0.4 ^e^	138.7 ± 7.2 ^d^	80.6 ± 0.8 ^b,c^	82.2 ± 0.8 ^c^	69.9 ± 2.5 ^a^	75.7 ± 1.1 ^b^

^a–g^, Mean values with different letters in the same row are statistically different (*p* ≤ 0.05) (*n* = 4). ^1^ Expressed as mg/kg. ^2^ Expressed as mmol/kg. * Sum of hydroxytyrosol, tyrosol and their secoiridoid derivatives.

**Table 2 antioxidants-11-00539-t002:** Linear Arrhenius kinetics of simple phenols and α-tocopherol fitted to a pseudo-zero-order reaction: *k* = A∙e ^₋Ea/RT^.

	Hydroxytyrosol		Tyrosol		α-Tocopherol	
	*Ea* (kJ/mol)	*A (*week^−1^*)*	*Ea* (kJ/mol)	*A (*week^−1^*)*	*Ea* (kJ/mol)	*A (*week^−1^*)*
I	81.7 ± 13.8 ^a^	9.4∙10^+10^ ± 1.9∙10^+02 a^	43.3 ± 15.2 ^a,b^	3.0∙10^+04^ ± 3.3∙10^+02 a^	85.3 ± 5.3 ^a^	1.4∙10^+12^ ± 7.6 ^b^
II	49.7 ± 8.4 ^a^	2.3∙10^+05^ ± 2.5∙10^+01 b^	91.1 ± 37.7 ^b^	3.4∙10^+12^ ± 1.7∙10+^06 b^	93.4 ± 10.7 ^a^	1.6∙10^+13^ ± 5.8∙10^+01 a^
III	84.6 ± 46.5 ^a^	1.2∙10^+11^ ± 4.8∙10^+07 c^	66.5 ± 11.3 ^a,b^	2.1∙10^+08^ ± 7.5∙10^+01 c^	88.4 ± 9.9 ^a^	3.8∙10^+12^ ± 4.4∙10^+01 c^
IV	64.2 ± 21 ^a^	5.4∙10^+07^ ± 2.9∙10^+03 d^	57.0 ± 12.2 ^a,b^	6.4∙10^+06^ ± 1.0∙10^+02 d^	92.5 ± 6.8 ^a^	1.6∙10^+13^ ± 1.3∙10^+01 a^
V	73.3 ± 6.5 ^a^	3.7∙10^+09^ ± 1.2∙10^+01 e^	27.4 ± 18.5 ^a^	3.5∙10^+01^ ± 1.1∙10^+03 a^	92.0 ± 7.8 ^a^	1.3∙10^+13^ ± 1.9∙10^+01 d^
VI	71.2 ± 4.8 ^a^	4.7∙10^+09^ ± 6.2 ^f^	55.9 ± 8.7 ^a,b^	1.3∙10^+07^ ± 2.7∙10^+01 e^	85.1 ± 14.9 ^a^	7.0∙10^+11^ ± 2.9∙10^+02 e^
VII	80.0 ± 14.7 ^a^	5.8∙10^+10^ ± 2.7∙10^+02 g^	49.6 ± 14.9 ^a,b^	1.7∙10^+05^ ± 2.8∙10^+02 a^	92.9 ± 12.1 ^a^	1.3∙10^+13^ ± 1.0∙10^+02 f^
	0.623 ≤ R^2^ ≤ 0.991		0.5240 ≤ R^2^ ≤ 0.954		0.942 ≤ R^2^ ≤ 0.992	

^a–g^, Same letters in the same row indicate no significant differences between samples *p* < 0.05 (*n* = 4).

**Table 3 antioxidants-11-00539-t003:** Linear Arrhenius kinetics of of complex phenols fitted to a pseudo-first-order reaction: *k* = A∙e ^₋Ea/RT^.

	o-Diphenols		Total Phenols		Sec. Hydroxytyrosol		Sec. Tyrosol	
	Ea (kJ/mol)	A (week^−1^)	Ea (kJ/mol)	A (week^−1^)	Ea (kJ/mol)	A (week^−1^)	Ea (kJ/mol)	A (week^−1^)
I	66.9 ± 3.4 ^a,b^	4.9∙10^+09^ ± 4 ^a^	58.6 ± 7.8 ^a^	1.4∙10^+08^ ± 19 ^a^	65.7 ± 3.5 ^a,b^	3.6∙10^+09^ ± 4 ^a^	53.79 ± 14.1 ^a^	1.9∙10^+07^ ± 220 ^a^
II	59.4 ± 5.6 ^a^	2.5∙10^+08^ ± 8 ^b^	52.0 ± 11.3 ^a^	1.1∙10^+07^ ± 73 ^b^	60.7 ± 6.0 ^a^	4.8∙10^+08^ ± 10 ^b^	41.26 ± 7.9 ^a^	1.5∙10^+05^ ± 20 ^b^
III	58.6 ± 5.9 ^a^	3.1∙10^+08^ ± 9 ^b^	53.3 ± 8.2 ^a^	2.7∙10^+07^ ± 23 ^c^	61.2 ± 7.1 ^a^	1.0∙10^+09^ ± 15 ^c^	44.04 ± 3.6 ^a^	7.2∙10^+05^ ± 4 ^c^
IV	74.6 ± 7.1 ^a,b,c^	1.4∙10+^11^ ± 15 ^c^	56.1 ± 10.1 ^a^	6.1∙10^+07^ ± 46 ^d^	68.7 ± 6.3 ^a,b^	2.0∙10^+10^ ± 11 ^d^	56.85 ± 12.3 ^a^	7.6∙10^+07^ ± 110 ^d^
V	71.5 ± 6.1 ^a,b,c^	3.8∙10^+10^ ± 10 ^d^	51.3 ± 7.8 ^a^	9.8∙10^+06^ ± 19 ^e^	62.0 ± 5.3 ^a^	1.4∙10^+09^ ± 7 ^e^	43.50 ± 12.3 ^a^	4.6∙10^+05^ ± 110 ^e^
VI	77.0 ± 3.1 ^b,c^	3.2∙10^+11^ ± 3 ^e^	63.4 ± 2.5 ^a^	8.4∙10^+08^ ± 3 ^f^	63.4 ± 2.0 ^a^	2.2∙10^+09^ ± 2 ^f^	59.36 ± 4.7 ^a^	2.4∙10^+08^ ± 6 ^f^
VII	84.2 ± 8.4 ^c^	5.2∙10^+12^ ± 24 ^f^	67.8 ± 2.4 ^a^	4.5∙10^+09^ ± 3 ^g^	73.9 ± 6.1 ^b^	1.1∙10^+11^ ± 10 ^g^	55.74 ± 3.9 ^a^	3.6∙10^+07^ ± 4 ^g^
	0.980 ≤ R^2^ ≤ 0.997		0.914 ≤ R^2^ ≤ 0.998		0.981 ≤ R^2^ ≤ 0.998		0.862 ≤ R^2^ ≤ 0.991	

^a–g^, Same letters in the same row indicate no significant differences between samples *p* < 0.05 (*n* = 4).

**Table 4 antioxidants-11-00539-t004:** Variables used in PCAs.

Variables Related to Initial State		Variables Related to Oxidation Progress	
Abbreviations	Meaning	Abbreviations	Meaning
CAROT	Carotens initial content	kPV	Rate constant of peroxide value
CHLOR	Chlorophyls initial content	kK_232_	Rate constant of K232
HTyr	Hydroxytyrosol initial content	kK_270_	Rate constant of K270
Tyr	Tyrosol initial content	kTPh	Rate constant of total phenols
TPh	Total Phenols initial content	kodPh	Rate constant of o-diphenols
secHTyr	Hydroxytyrosol secoiridois initial content	ksecHTyr	Rate constant of Hydroxytyrosol secoiridois
secTyr	Tyrosol secoiridois initial content	kαToh	Rate constant of α-tocopherol
odPh	o-diphenols initial content	kUFAs	Rate constant of Unsaturated Fatty Acids
Tyr + Der	Tyrosol + Tyrosol secoiridois initial content	kPUFAs	Rate constant of Polyunsaturated Fatty Acids
αToh	α-tocopherol initial content	tPV	Time to reach PV = 20 meq/kg
UFAs	Unsaturated Fatty Acids initial content	tK_232_	Time to reach K232 = 2.50
PUFAs	Polyunsaturated Fatty Acids initial content	tK_270_	Time to reach K270 = 0.22
PV	Initial Peroxide Value	T	Temperature of the oxidation experiment
K232	Initial K232		
K270	Initial K270		
RancimatOS	Rancimat Oxidative Stability		

**Table 5 antioxidants-11-00539-t005:** Components, loadings and scores of PCA1.

Component Matrix ^a^ (Loadings)				Rotated Component Matrix ^a^ (Loadings)			Components Score Coeficient Matrix		
	Component			Component			Component		
	1	2	3	1: Non Oxidisable Substrate	2: Initial Oxidation State and Conditions	3: Simple Phenols	1: Non Oxidisable Substrate	2: Initial Oxidation State and Conditions	3: Simple Phenols
T	−0.183	0.538	0.627	−0.178	0.822	−0.094	−0.020	0.183	0.423
PV	−0.341	0.720	0.257	−0.295	0.716	0.318	−0.038	0.246	0.173
K232	0.152	0.731	0.274	0.195	0.725	0.261	0.017	0.249	0.185
K270	0.529	0.539	0.327	0.550	0.610	0.055	0.059	0.184	0.221
CAROT	0.864	0.052	−0.002	0.863	0.021	−0.055	0.096	0.018	−0.001
CHLOR	0.778	0.105	0.057	0.778	0.101	−0.055	0.086	0.036	0.038
HTyr	0.015	0.762	−0.515	0.113	0.229	0.884	0.002	0.260	−0.347
Tyr	−0.312	0.691	−0.635	−0.210	0.102	0.961	−0.035	0.236	−0.429
TPh	0.968	0.010	−0.148	0.973	−0.110	0.015	0.107	0.003	−0.100
secHTyr	0.974	−0.138	−0.040	0.959	−0.149	−0.163	0.108	−0.047	−0.027
secTyr	0.933	−0.190	0.069	0.907	−0.115	−0.274	0.103	−0.065	0.046
odPh	0.973	−0.049	−0.102	0.970	−0.124	−0.058	0.108	−0.017	−0.069
Tyr + Der	0.919	0.106	−0.218	0.937	−0.084	0.136	0.102	0.036	−0.147
αToh	0.726	−0.147	0.283	0.691	0.063	−0.384	0.080	−0.050	0.191
UFAs	−0.686	−0.297	−0.020	−0.706	−0.222	−0.108	−0.076	−0.101	−0.014
PUFAs	−0.832	−0.221	0.047	−0.849	−0.118	−0.093	−0.092	−0.075	0.031
RancimatOS	0.945	−0.005	−0.008	0.940	−0.028	−0.096	0.105	−0.002	−0.006
Extraction method: Principal Components Analysis				Rotation method: Varimax. Highest loadings in grey color			Rotation method: Without rotation Components score		

^a^ Rotation converged in 5 iterations

**Table 6 antioxidants-11-00539-t006:** Proposed predictive models.

Dependent Variable	Models by Stepwise MLR	R^2^
tPV_25	42.56 + 17.85 degradation rates −16.57 initial oxidation state and conditions −14.20 free simple phenols +8.51 non-oxidizable substrate	0.575
tK_232__25	20.15 +7.19 degradation rates −8.74 initial oxidation state and conditions −9.53 free simple phenols +5.51 non-oxidizable substrate	0.501
kodPh_25	−0.095 + 0.082 degradation rates + 0.011 non-oxidizable substrate	0.937
ksecHTyr_25	−0.106 + 0.087 degradation rates +0.006 non-oxidizable substrate +0.004 initial oxidation state and conditions	0.969
tPV_25	71.78+ 4.92 tK_232__40 + 1211.12 kodPh_40+ 8.47 tPV_60	0.881
tK_232__25	6.23 + 7.63 tK_232__40 + 490.05 kodPh_40	0.904
kodPh_25	−0.008 −0.005 kPV_50 + 0.001PV − 0.001 tK_232__50	0.901
ksecHTyr_25	−0.029 + 0.114 K_270_ + 8.104 × 10^−5^ PUFAs + 0.111 kTPh_50 + 0.004 kUFAs_50	0.915

## Data Availability

Data are available in the manuscript and Appendix A.

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
