# Peer review of "Modelling Virgin Olive Oil Potential Shelf-Life from Antioxidants and Lipid Oxidation Progress"

_antioxidants, 2022, doi:10.3390/antiox11030539_

Round 1

Reviewer 1 Report

Virgin olive oil is an important ingredient for food preparation. Nevertheless, this oil is sensitive to oxidation processes, which decrease dramatically the oil quality. Therefore, the topic of this manuscript, which deals with shelf life and effects of antioxidant constituents on the oxidation rate, is interesting and important.

The authors investigated the rate of oxidation reactions in virgin olive oil samples in various temperature reaction conditions. Additionally, they explored effects on the antioxidant on the reaction rate. For this kinetic study, the traditional Arrhenius method was employed.

The strengths of this manuscript are well-planned set of experiments and adequate well-established methods use. Moreover, the used methods are well-described.

The results are supported by the experimental data, which are presented by illustrative graphs and tables. Furthermore, the results are properly interpreted and discussed. The manuscript is logically arranged and shows a high professionality of the author team. Since I found no serious weaknesses in the manuscript, I recommend the paper for publication.

Author Response

The author greatly thanks Reviewer 1's valuation and comments.

Reviewer 2 Report

The manuscript by Mancebo-Campos et al. reports a valuable model that explains the correlation between virgin olive oil oxidation parameters, initial composition, and progressive degradation of major and minor compounds. Noteworthy, considering that the EFSA health claim “olive oil polyphenols protect LDL particles from oxidative damage” may be used for olive oil that contains at least 5 mg of hydroxytyrosol and its derivatives per 20 g of olive oil; one of the proposed models can be used to estimate the moment the concentration would fall off below 5 mg/20 g of olive oil simply by knowing the initial content of hydroxytyrosol derivatives. Finally, some typos here and there in the text should be corrected, and always the same acronym to indicate extra virgin olive oil (VOO or EVOO) should be adopted.

Author Response

The author greatly thanks Reviewer 2's valuation and comments. The used of the acronym EVOO has been changed to VOO. The full text has been revised and some typos were corrected.

Reviewer 3 Report

  1. The authors claimed an “accelerated shelf-life test at mild temperature (40–60 °C)”, while in “2.2. Oxidation experiments”, the authors described “25, 40, 50 and 60 °C during 93, 41, 34 and 19 weeks”. In my opinion, some of them are not mild conditions (and not shelf-life conditions), and the authors did not mention any related evaluation standards or rules. So, please provide detailly the reasons/basis to design such conditions.

  1. Did the author use any methods to control or determine the oxygen concentration in the experimental environment?

  1. Please report the experiment duplicates (n = ?). If the model is sensitive to the unexpected outliner, the simulation may be bothered by deviated data.

  1. May the authors explain why different samples showed different behaviors (Fig.1 ~ Fig.3) from the view of chemical composition difference? Moreover, can the authors summarize the most significant (the key) compound(s) that decide the lipid oxidation behavior?

  1. 3: According to the authors, α-tocopherol showed more variation among different temperatures than hydroxytyrosol and o-diphenol. Is there any possible explanation from the view of “antioxidation capacity” or “ease of being oxidized” among these constituents? Besides, for α-tocopherol, is there any separator (“watershed”) of temperature? Not only temperature- but also time-dependent dynamic changes are helpful for a complete and comprehensive investigation.

  1. Although there are tables for listing the variables (Table 4) and loadings/scores (Table 5), there is no plotted data (figure). For example, Table 5 could be better expressed as a score-loading-biplot.

  1. Are there any overfitting and collinearity issues for these models? Are there any validations by using either another dataset or cross-validation?

Author Response

We greatly appreciate your valuable feedback. Please find attached the answers to your questions and how the paper includes your suggestions. 

Round 2

Reviewer 3 Report

The authors have responded to the comments and made proper modifications. And they corrected some mistakes to make the draft a better quality. Therefore, I have no more questions for this manuscript.